# Association of Carbohydrate and Fat Intake with Prevalence of Metabolic Syndrome Can Be Modified by Physical Activity and Physical Environment in Ecuadorian Adults: The ENSANUT-ECU Study

**DOI:** 10.3390/nu13061834

**Published:** 2021-05-27

**Authors:** Christian F. Juna, Yoonhee Cho, Dongwoo Ham, Hyojee Joung

**Affiliations:** 1Department of Public Health, Graduate School of Public Health, Seoul National University, Seoul 08826, Korea; christian@snu.ac.kr; 2Facultad de Enfermería, Pontificia Universidad Católica del Ecuador, Quito 170525, Ecuador; 3Department of Biomedical and Pharmaceutical Sciences, The University of Montana, Missoula, MT 59812, USA; 4Institute of Health and Environment, Seoul National University, Seoul 08826, Korea; dwhampch@snu.ac.kr

**Keywords:** carbohydrate and fat intake, elevation, humidity, metabolic syndrome, ENSANUT-ECU, physical inactivity

## Abstract

The associations of lifestyle and environment with metabolic syndrome (MetS) and cardiovascular disease have recently resulted in increased attention in research. This study aimed to examine interactive associations among carbohydrate and fat intake, physical environment (i.e., elevation and humidity), lifestyle, and MetS among Ecuadorian adults. We used data from the Ecuador National Health and Nutrition Survey 2012 (ENSANUT-ECU), with a total of 6023 participants aged 20 to 60 years included in this study. Logistic regression was used to determine the association of status of carbohydrate and fat intake, low-carbohydrate high-fat diet (LCHF) and medium-carbohydrate and fat (MCF) diet with MetS, where the high-carbohydrate low-fat (HCLF) diet was used as a reference. Women with LCHF and MCF diets showed lower prevalence of increased blood pressure (OR = 0.34, 95% CI: 0.19–0.59; OR = 0.50, 95% CI: 0.32–0.79, respectively). Women with MCF diet also showed lower prevalence of elevated fasting glucose (OR = 0.58, 95% CI: 0.37–0.91). Moreover, there were negative associations between MetS and reduced HDL cholesterol in women with MCF diet residing in low relative humidity (OR = 0.66, 95% CI: 0.45–0.98) and in women with LCHF diet residing at a high elevation (OR = 0.37, 95% CI: 0.16–0.86). Additionally, higher prevalence of increased waist circumference was observed in men with both MFC and LCHF diets who were physically inactive (OR = 1.89, 95% CI: 1.12–3.20; OR = 2.34, 95% CI: 1.19–4.60, respectively) and residing in high relative humidity (OR = 1.90, 95% CI: 1.08–2.89; OR = 2.63, 95% CI: 1.32–5.28, respectively). Our findings suggest that LCHF intake is associated with lower blood pressure, while MCF intake is associated with lower blood pressure and fasting glucose in Ecuadorian women. Furthermore, the associations of carbohydrate and fat intake with prevalence of MetS can be modified by physical activity, relative humidity, and elevation. The obtained outcomes may provide useful information for health programs focusing on dietary intake and lifestyle according to physical environment of the population to promote health and prevent metabolic diseases.

## 1. Introduction

Metabolic syndrome (MetS), characterized by the presence of hyperglycemia, hypertension, hypertriglyceridemia, low high-density lipoprotein cholesterol (HDL), and abdominal obesity, is a major public health problem [1]. Over a billion people worldwide are estimated to have MetS [2], and in most developing countries, MetS represents one of the largest contributors to the global burden of type 2 diabetes mellitus (DM2) and cardiovascular disease (CVD) [3,4]. Like to other Latin American countries, Ecuador reports a 31.2% prevalence of MetS and 85% of the total population has at least one of its abnormalities [5]. In 2019, DM2 and CVD constituted the first causes of mortality in Ecuadorian adults (6.5% and 6.2%, respectively) [6]. Thus, it is urgent to identify potential determinants of MetS development in Ecuador for better strategies to prevent MetS onset, as well as its subsequent diseases.

Complex interactions between genetics, physical environment (i.e., elevation and humidity), and health-related lifestyles have been identified to lead to onset of MetS, though exact causes remain unknown [7,8,9,10]. In reference to environmental and lifestyle factors, low elevation (<2000 masl), high relative humidity (>80%), inadequate food intake, alcohol consumption, tobacco smoking, and physical inactivity have been reported as risk factors for MetS development [7,8,11,12,13]. However, an extensive investigation between dietary intake and MetS has not yet been made among Ecuadorians, and most data in healthy population come from research in high-income countries, where findings on this topic are yet controversial [14,15,16,17].

Recent scientific evidence on health benefits of low-carbohydrate high-fat (LCHF) diets focusing on weight loss and decrease of risk of CVD has made many people intentionally change their carbohydrate and fat intake patterns. Several meta-analyses of randomized controlled clinical trials (RCTs) revealed that a low-carbohydrate diet resulted in reduction of body weight and positive changes in triglycerides and high-density lipoprotein (HDL) levels, but increased low-density lipoprotein (LDL) levels [17,18]. However, another meta-analysis of RCTs showed that weight loss, blood pressure, or lipid profile were not significantly different in populations with low-carbohydrate and energy balanced diets [19], which suggest the effect of low-carbohydrate diet on MetS and/or CVD is still controversial. Furthermore, most of these investigations focusing on dietary patterns and MetS involved adults from the US and European countries, and a few studies were conducted in Latin American populations [20,21]. One study on the Brazilian population found that lower protein and higher carbohydrate intakes were related to lower HDL levels and higher risk of hypertriglyceridemia in women [22].

These worldwide findings indicate that the associations between LCHF diet and metabolic disorders may fluctuate in terms of sociodemographic characteristics, lifestyles, and environmental conditions of the populations; thus, there is a need to investigate interactive associations among various dietary factors and environmental conditions in various populations. In this context, we aimed to examine the associations between carbohydrate and fat intake and MetS according to health-related lifestyles and physical environment among Ecuadorian adults.

## 2. Materials and Methods

### 2.1. Study Design

The present study used data from the Ecuador National Health and Nutrition Survey (ENSANUT-ECU) 2012, a nation-wide cross-sectional survey conducted by Ecuador’s Ministry of Public Health. The ENSANUT-ECU collected information on sociodemographic status, health-risk and food consumption, and anthropometry, blood pressure, and blood biomarkers. Detailed explanations of ENSANUT-ECU are available elsewhere [23].

### 2.2. Subjects

Participants in this study were Ecuadorian adults from ages 20 to 60 years who completed a dietary recall survey as well as anthropometric and biochemical measurements which related to MetS (11,044 participants). Participants with missing information on physical activity (*n* = 4166) and lifestyles (*n* = 494), or participants who took medication for hypertension and other diseases (*n* = 361) were excluded. Thus, the final 6023 Ecuadorian adults (1964 men and 4059 women) were selected for this study.

### 2.3. General Characteristics

The sociodemographic data of participants including sex, age, ethnicity, education, and family economic status were obtained using the standardized questionnaire for housing information. Ethnicity was classified as mestizo (Amerindian with European mix) and others (i.e., Amerindian, montubio, Afro-Ecuadorian, and white); education as primary (≤7th grade), secondary (>7th grade–≤ 12th grade), and college or higher; and family economic status was categorized as low (first and second quintile), middle (third and fourth quintiles), or high (fifth quintile).

### 2.4. Physical Environmental Conditions

Information on elevation of residence was collected in the housing questionnaire of ENSANUT. It was obtained through georeferencing; elevation was divided in two groups, high elevation for residence over 2000 masl and low elevation for residence under 2000 masl [24,25]. The data on relative humidity (%) in 2012 of the participant’s residence was obtained from the National Institute for Meteorology and Hydrology [26]. According to previous studies, high relative humidity was established if residence was above 80% and low relative humidity below 80% [27,28].

### 2.5. Health-Related Lifestyles

According to the United States National Survey on Drug Use and Health [29] and the ENSANUT-ECU, current alcohol consumption and smoking were determined as “yes” if the participant had drunk alcoholic beverages and smoked in the past 30 days of data collection. Physical activity was defined as “yes” if the participant had performed vigorous-intensity activity for at least 75 min, moderate-intensity activity for at least 150 min, or both for the past 7 days prior to the collection of data, and “no” for any activity that that took less than 75 min [30].

### 2.6. Dietary Assessment

Participants’ intake of macronutrients (such as carbohydrate, protein, and fat) were estimated from the information on all foods and beverages they consumed a day collected by 24-h recall method. Macronutrient intake was calculated using the food composition table of the Food Dietary Guidelines for the Ecuadorian population (GABA) [31]. Participants were divided into three groups based on their macronutrient intake data: LCHF (energy from carbohydrate <45% and >30% from fat), HCLF (energy from carbohydrate >65% and <20% from fat), and MCF (45% to 65% of energy from carbohydrate and 20% to 30% from fat) [18]. The adequacy of energy intake was evaluated through percentages of estimated energy requirements (EER), in accordance with the age-, sex-, weight-, height- and physical activity level-specific equations [32,33].

### 2.7. Metabolic Syndrome

Participants’ height, weight, waist circumference, and blood pressure were measured twice at their residences by trained technicians using standardized procedures [34]. The mean of the readings was used for this study. Body mass index (BMI) was calculated from weight and height data (kg/m^2^). Fasting glucose, total cholesterol, HDL cholesterol, and triglyceride levels were measured from blood samples of participants who fasted for at least 8 h before examination. Blood samples were measured using an enzymatic-colorimetric assay Modular Evo-800 (Roche Diagnostics) and Friedewald’s formula was used to calculate LDL cholesterol [35]. The detailed laboratory procedures of ENSANUT-ECU are available elsewhere [23].

The subjects with MetS were identified when they had three or more of the following components: (1) waist circumference (men ≥ 94 cm, women ≥ 88), (2) blood pressure (systolic ≥ 130 mmHg and/or diastolic ≥ 85 mmHg), (3) HDL cholesterol (men < 40 mg/dL and women < 50 mg/dL), (4) elevated triglycerides (≥50 mg/dL), and (5) elevated fasting glucose (≥100 mg/dL). The diagnosis criteria were based on the National Cholesterol Education Program Adult Treatment Panel III and the Latin American Diabetes Association [36,37].

### 2.8. Statistical Analyses

All analyses were stratified by sex and status of carbohydrate and fat intake (LCHF, HCLF, MCF). All continuous variables were presented as means with standard errors (SE), and categorical variables were presented as percentages. Differences in groups were compared using chi-square and ANOVA depending on the variable analyzed. The adjusted means and SE of the anthropometric and biochemical variables according to the diet type were estimated using a generalized linear model after adjusting for ethnicity, family economic status, education level, elevation, BMI (except for the model of waist circumference), and total energy intake. Odds ratio (ORs) and 95% confidence intervals (CIs) for MetS according to diet type were estimated using multiple logistic regression analysis after adjusting for confounding variables, including ethnicity, family economic status, education level, elevation, relative humidity, BMI (except for the model of waist circumference), and total energy intake. Statistical analyses were performed using Statistical Analysis Systems (SAS) software version 9.4 [38] applying the PROC SURVEY procedure. All *p*-values were two-tailed and a *p*-value of <0.05 was considered statistically significant.

## 3. Results

Table 1 shows the general characteristics of the study participants by sex and carbohydrate and fat intake based on carbohydrate and fat content. The mean ages were 34.6 ± 0.44 (SE) years for men and 35.2 ± 0.35 years for women. Household income, education level, and elevation of residence were significantly different from the diet groups in both sexes (*p* < 0.05). However, relative humidity was significantly different in women only (*p* = 0.0004).

Table 2 shows anthropometric and biochemical measurements for MetS components and macronutrient intakes. Blood pressure, total cholesterol, triglycerides, and energy intake were higher in men, but BMI and EER (%) were higher in women. Significant differences between diet groups were observed in BMI, systolic blood pressure, diastolic blood pressure, and HDL cholesterol in both sexes (*p* < 0.05), while in men, waist circumference was significantly different (*p* < 0.0001) and in women, fasting glucose, total cholesterol, LDL cholesterol and triglycerides (*p* = 0.0126, *p* = 0.0002, *p* < 0.0001, *p* = 0.0008, respectively). Moreover, men and women with HCLF diets showed a higher energy intake and EER (%) (*p* < 0.001, both). Mean contribution rate to energy of carbohydrate, protein, and fat comprised 42.2%, 15.6%, 38.3% in men with LCHF, and 40.7%, 15.9%, 39.3% in women with LCHF.

The prevalence and ORs for MetS according to carbohydrate and fat intake are summarized in Table 3. Women with LCHF and MCF diet groups showed lower prevalence of elevated blood pressure (10.7%, OR = 0.34, 95% CI: 0.19–0.59; 13.7%, OR = 0.50; 95% CI: 0.32–0.79, respectively), and women with MCF diet showed a lower prevalence of elevated fasting glucose (OR = 0.58, 95% CI: 0.37–0.91) after adjusting for confounding variables.

Further analyses for associations between diet and MetS according to health-related lifestyles and physical environment (i.e., elevation and humidity) were performed to explore potential interactions among the variables. When compared to participants with HCLF diets, men with LCHF and MFC diet who were physically inactive had a higher prevalence of increased waist circumference (OR = 2.34, 95% CI: 1.19–4.60; OR = 1.89, 95% CI: 1.12–3.20; respectively) (Figure 1A) and living in high relative humidity (OR = 2.63, 95%CI:1.32–5.28; OR = 1.90, 95%CI:1.08–2.89; respectively) (Figure 1B), whereas men with MCF diet performing exercise showed a lower prevalence of elevated triglycerides (OR = 0.45, 95% CI: 0.21–0.98) as shown in Figure 1C.

In addition, women with an LCHF diet living at high elevation showed a significantly lower prevalence of reduced HDL cholesterol (OR = 0.37, 95% CI: 0.16–0.86) (Figure 2A), while women with MCF intake living in low relative humidity showed a lower prevalence of MetS (OR = 0.66, 95% CI: 0.45–0.98) (Figure 2B).

## 4. Discussion

In this study, we investigated the associations of carbohydrate and fat intake with MetS according to health-related lifestyles and physical environment (i.e., elevation and humidity) in the Ecuadorian adult population, and found that LCHF and MCF diets were inversely associated with elevated blood pressure in women, while women with MCF diet showed lower prevalence of elevated fasting glucose. Moreover, there were negative associations between MetS and reduced HDL cholesterol in women with MCF diet residing in low relative humidity and in women with LCHF diet residing at a high elevation, whereas higher prevalence of increased waist circumference was observed in men with both MFC and LCHF diets who were physically inactive and residing in high relative humidity. To the best of our knowledge, this is the first study using a large population in Ecuador to explore the interactive associations of carbohydrate and fat intakes, lifestyles, and physical environment with MetS.

The Ecuadorian diet is generally high in carbohydrate and fat [23]. Although there is no a universal definition of low-carbohydrate and high-fat diet, several studies agree that in Western populations a low-carbohydrate high-fat diet consists of less than 45% of carbohydrates [19,39], and more than 30% of fats [40] as sources of energy. The intake of LCHF and MCF diets has been found to attenuate blood pressure [41,42]; consequently, these facts could be explained by the reduction of HDL cholesterol and triglycerides levels, known as a major risk factors for cardiovascular events and hypertension [43]. Moreover, a LCHF diet may induce hypoglycemia and decrease oxidative stress, which could cause endothelial vasoconstriction and therefore decrease blood pressure [44]. Conversely, other studies did not find significant associations among proportions of carbohydrate and fat intake and hypertension [45]. In addition, several RCTs have reported that the greater the carbohydrate restriction, the greater the glucose-lowering effect [46], which is in accordance with our findings. The association of carbohydrate and fat intake with glucose levels may be explained by the hyperinsulinemia and postprandial hyperglycemia caused by a decrease in carbohydrate intake [47]. On the contrary, HCLF diet has been associated with a higher prevalence of MetS [48] and reduced HDL cholesterol [49]. The associations between LCHF, MCF, and HCLF are still unclear. Thus, further studies are needed to clarify the optimal types and proportions of carbohydrate and fat and the mechanisms underlying their associations.

In addition, protein content exhibited significant differences among diet groups in this study; an increased intake of protein has been reported to have a greater satiety effect than intake of carbohydrates, helping control hunger between meals and regulate the processes of energy expenditure, thermogenesis and glucose metabolism, which intervene in the pathophysiology of MetS [50,51]. The association between protein intake and MetS is still unclear, as some studies have not found associations [52,53,54] while others have found that low protein intake could be a risk factor for MetS components [22].

To further explore our findings, we analyzed the interactive associations of MetS with carbohydrate and fat intake, physical environment, and health-related lifestyles. Ecuador is a megadiverse country composed by four different geographical regions (the Coast, the Galapagos Islands, the Amazon, and the Andean region); its population resides in a wide elevation range of 0 masl to 3900 masl. In addition, Ecuador has 11 different types of microclimates ranging from Tropical to Oceanic and is located on the equatorial line, thus producing little seasonality. An adequate macronutrient intake was inversely associated with MetS in women living in low humidity, and a lower prevalence of reduced HDL cholesterol was found in women residing at high elevation. Several studies investigated the effects of physical environment on human’s health and found that high relative humidity is associated with coronary diseases [55,56], DM2 [27], and MetS [10]. Moreover, epidemiological studies have described that susceptibility to extreme humidity and temperature varies by gender [57], which supports our findings. While the associations of high relative humidity with health outcomes are still controversial, our results showed that high relative humidity combined with LCHF and/or MCF diets may trigger MetS. In the case of elevation, some studies have reported an inverse association between high elevation and MetS and its components, which may be explained by physiological adaptations to hypoxia that increase hemoglobin concentrations and accelerate glucose tolerance and the metabolism of lipids [58,59,60,61,62,63]. Moreover, exposure to high elevation has been found to decrease the levels of HDL, LDL, and associated proteins due to maturation of lipoprotein particles [64].

In addition, we found that physically inactive men and men living in high relative humidity showed a higher prevalence of abdominal obesity, whereas physically active men with an MCF intake and residing in low relative humidity showed a lower prevalence of elevated triglycerides. These findings may be explained by the associations of high relative humidity with heat stress and fatigue, which cause reduction in exercise capacity [65,66]. Additionally, some studies have shown a positive impact of exercise on blood cholesterol and triglyceride profiles, abdominal obesity, and DM2, likely due to insulin resistance, adipose and protein metabolism, and epigenetic factors [67,68]. Thus, our findings should be interpreted with caution.

This study has several limitations. First, the cross-sectional design of the ENSANUT-ECU data made it difficult to identify the causal relationship between carbohydrate and fat intake with MetS. Further prospective studies are required to examine the effect of these diets on MetS in Ecuadorian adults. Second, data on dietary intake from the 24 h dietary recall might not represent the usual food intake of participants. However, this study included a nation-wide representative sample of participants. Third, we could not estimate the differences in carbohydrate quality and fatty acid composition in diet groups due to the lack of information. Therefore, more studies are needed to clarify their associations with MetS and its components. Despite these limitations, to the best of our knowledge, this is the first study based on a nationally representative sample of Ecuadorian adults that examined the interactive associations among carbohydrate and fat intake, health-related lifestyle, and physical environment with MetS and its abnormalities.

## 5. Conclusions

The obtained results suggest that LCHF and MFC compared to HCLF intakes may play protective roles in the onset of MetS, and their associations can be modified by physical activity and physical environment such as relative humidity and elevation in Ecuadorian adults. Thus, health programs focusing on dietary intake to promote health and prevent metabolic diseases should consider other factors, including health-related lifestyles and physical environment conditions of the population.

## Figures and Tables

**Figure 1 nutrients-13-01834-f001:**
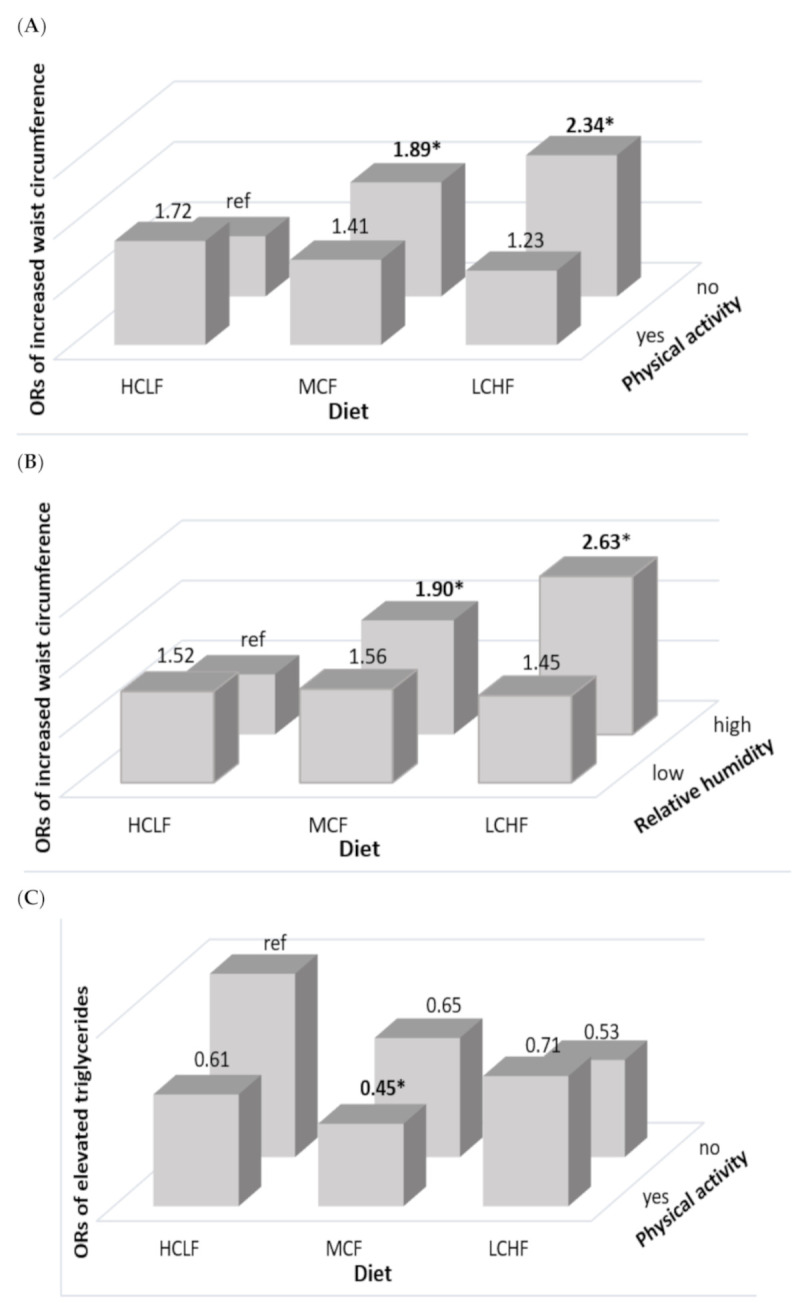
Adjusted odds ratios of MetS in Ecuadorian men according to diet, physical activity, and environment. (**A**) Odd ratios for increased waist circumference by the status of carbohydrate and fat intake and physical activity; (**B**) odd ratios for increased waist circumference by the status of carbohydrate and fat intake and relative humidity; and (**C**) odd ratios for elevated triglycerides by the status of carbohydrate and fat intake and physical activity. All values accounted for the complex sampling design effect of the national surveys using PROC SURVEY procedure. Ethnicity, family economic status, education level, elevation, relative humidity, BMI (except for the model of waist circumference), and total energy intake were adjusted for the multiple logistic regression. Yes or No for physical activity, and high (>80%) or low (50–80%) for relative humidity. * indicates statistical significance (*p* < 0.05).

**Figure 2 nutrients-13-01834-f002:**
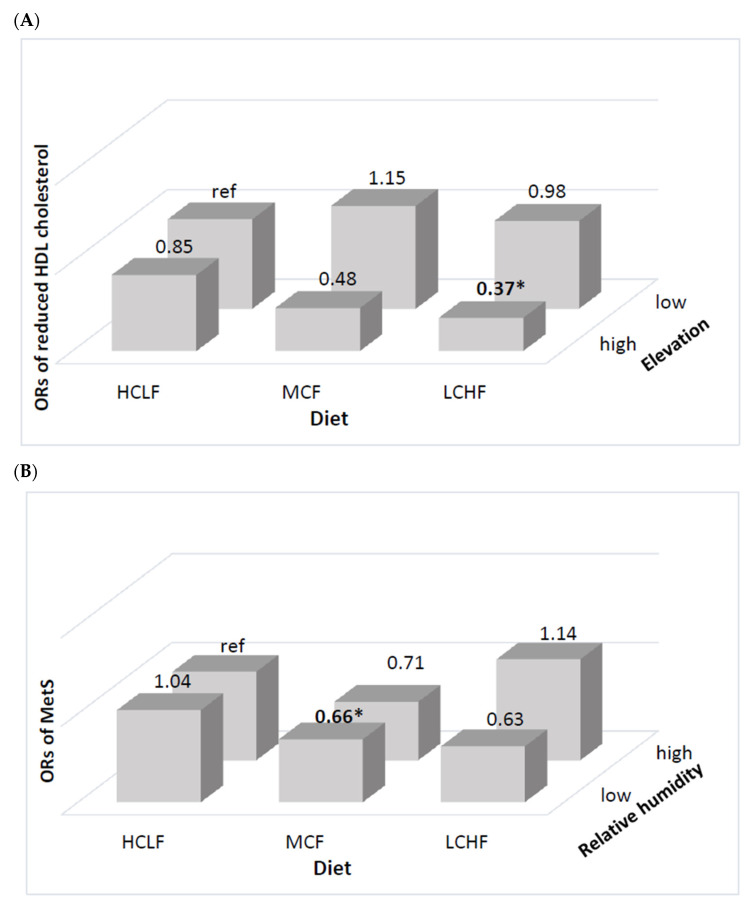
Multivariable-adjusted odds ratios for MetS in Ecuadorian women according to diet and physical environment. (**A**) Odd ratios for reduced HDL cholesterol by the status of carbohydrate and fat intake and elevation; and (**B**) odd ratios for MetS by the status of carbohydrate and relative humidity. All values accounted for the complex sampling design effect of the national surveys using PROC SURVEY procedure. Ethnicity, family economic status, education level, elevation, relative humidity, BMI (except for the model of waist circumference), and total energy intake were adjusted for the multiple logistic regression. High (>2001 masl) or low (≤2000) for elevation and high (>80%) or low (50–80%) for relative humidity. * indicates statistical significance (*p* < 0.05).

**Table 1 nutrients-13-01834-t001:** Demographic, health-related lifestyle, and physical environmental characteristics of study participants by carbohydrate and fat intake.

	Men				Women			
Variables	LCHF	MCF	HCLF	*p*-Value ^a^	LCHF	MCF	HCLF	*p*-Value
Number (%)	330(16.8)	1410 (71.8)	224 (11.4)		797 (19.7)	2907 (71.6)	355 (8.7)	
Age(years), N (%)				0.7632				0.1004
20–29	121 (39.7)	514 (39.2)	70 (33.9)		255 (32.4)	1051 (37.1)	115 (33.7)	
30–39	106 (27.6)	468 (28.5)	85 (36.0)		293 (32.9)	1024 (28.9)	139 (33.0)	
40–49	80 (21.4)	317 (20.6)	50 (17.8)		222 (27.3)	691 (23.2)	79 (19.3)	
50–59	23 (11.3)	111 (11.7)	19 (12.3)		27 (7.4)	141 (10.8)	22 (14.0)	
Ethnicity, N (%)				0.0917				0.0486
Mestizo	304 (88.9)	1247 (82.3)	188 (79.4)		723 (87.2)	2562 (83.4)	298 (77.5)	
Others	26 (11.1)	163 (17.7)	36 (20.6)		74 (12.8)	345 (16.6)	57 (22.5)	
Family economic status ^b^, N (%)				<0.0001				<0.0001
Low	55 (11.7)	388 (26.7)	90 (35.7)		174 (18.1)	897 (26.8)	164 (45.9)	
Middle	133 (37.7)	695 (47.4)	105 (51.0)		350 (39.5)	1390 (45.5)	151 (42.9)	
High	142 (50.6)	327 (25.9)	29 (13.3)		273 (42.4)	630 (27.7)	40 (11.2)	
Education level, N (%)				<0.0001				<0.0001
Primary school	59 (18.9)	354 (23.4)	94 (43.0)		173 (21.8)	828 (27.2)	130 (41.4)	
Secondary school	147 (39.1)	714 (53.2)	103 (44.5)		369 (44.7)	1368 (47.2)	174 (46.1)	
College or higher	124 (42.0)	342 (23.4)	27 (12.5)		255 (33.5)	711 (25.6)	51 (12.5)	
Current alcohol consumption ^c^, N (%)				0.8775				0.4129
Yes	190 (58.7)	784 (57.8)	135 (55.7)		235 (27.2)	753 (25.6)	76 (21.7)	
No	140 (41.3)	626 (42.2)	89 (44.3)		562 (72.8)	2154 (74.4)	279 (78.3)	
Current smoking ^d^, N (%)				0.1374				0.1013
Yes	125 (33.3)	467 (29.6)	61 (22.6)		56 (7.1)	165 (6.7)	11 (2.4)	
No	205 (66.7)	943 (70.4)	163 (77.4)		741 (92.9)	2742 (93.3)	344 (97.6)	
Physical activity ^e^, N (%)				0.8032				0.5826
Yes	147 (44.0)	608 (42.0)	95 (44.6)		138 (19.2)	478 (17.7)	59 (15.5)	
No	183 (66.0)	802 (58.0)	129 (55.4)		659 (80.8)	2429 (82.3)	296 (84.5)	
Elevation ^f^, N (%)				<0.0001				<0.0001
High	167 (52.8)	557 (38.3)	59 (25.4)		367 (51.3)	1014 (38.0)	88 (26.5)	
Low	163 (47.2)	853 (61.7)	165 (74.6)		430 (48.7)	1893 (62.0)	267 (73.5)	
Humidity ^g^, N(%)				0.1431				0.0004
High	173 (30.5)	783 (34.5)	139 (59.0)		443 (30.6)	1843 (38.4)	249 (47.7)	
Low	157 (69.5)	627 (65.5)	85 (41.0)		354 (69.4)	1064 (61.6)	106 (52.6)	

**Abbreviations:** Low-Carbohydrate High-Fat (LCHF); Medium-Carbohydrate and Fat (MCF); High-Carbohydrate Low-Fat (HCLF) ^a^ Based on *χ*^2^ test for categorical variables and ANOVA for continuous variables. ^b^ Family economic status was categorized as low (first and second quintiles), middle (third and fourth quintiles), or high (fifth quintile). ^c^ Alcohol consumption was defined as “yes” for the consumption of alcoholic beverages over the past 30 days. ^d^ Current smoking was defined as “yes” for cigarette smoking over the past month. ^e^ Physical activity was defined as “yes” when performing vigorous activities for at least 75 min or moderate activities for at least 150 min over the past 7 days. ^f^ Elevation was defined as “high” residence was ≥ 2001 masl. ^g^ Humidity > 80% was classified as “high”.

**Table 2 nutrients-13-01834-t002:** Anthropometric and biochemical measurements of MetS and macronutrient intake of participants according to carbohydrate and fat intake.

Variables	Men				Women			
	LCHF	MCF	HCLF	*p*-Value ^a^	LCHF	MCF	HCLF	*p*-Value
	(N = 330)	(N = 1410)	(N = 224)		(N = 797)	(N = 2907)	(N = 355)	
Anthropometric and biochemical variables (mean ± SE)								
BMI (kg/m^2^)	26.3 ± 0.4	26.8 ± 0.2	26.1 ± 0.4	0.0433	27.1 ± 0.3	27.3 ± 0.1	28.1 ± 0.5	<0.0001
Waist circumference (cm)	101.9 ± 7.7	93.6 ± 1.2	93.7 ± 4.7	<0.0001	98.4 ± 5.1	94.9 ± 2.9	95.9 ± 3.9	0.1364
SBP (mmHg)	132.5 ± 7.9	122.3 ± 0.5	127.9 ± 1.7	<0.0001	116.3 ± 2.6	115.3 ± 0.9	120.1 ± 2.9	0.0007
DBP (mmHg)	88.8 ± 8.3	77.1 ± 0.6	81.9 ± 4.7	<0.0001	73.73 ± 2.7	72.6 ± 0.9	76.6 ± 2.9	0.0047
Fasting glucose (mg/dL)	94.4 ± 2.0	94.3 ± 1.3	94.0 ± 1.3	0.5412	93.0 ± 1.4	92.5 ± 0.8	95.3 ± 2.0	0.0126
Total cholesterol (mg/dL)	190.8 ± 3.5	184.8 ± 1.5	187.0 ± 3.8	0.5957	183.4 ± 2.0	179.7 ± 1.2	177.1 ± 2.8	0.0002
HDL cholesterol (mg/dL)	41.5 ± 1.1	40.8 ± 0.4	42.5 ± 0.9	0.0012	47.4 ± 0.7	46.3 ± 0.4	45.4 ± 0.9	<0.0001
LDL cholesterol (mg/dL)	113.2 ± 2.6	111.4 ± 1.3	110.4 ± 2.7	0.3310	112.2 ± 1.8	108.1 ± 0.9	106.6 ± 2.4	<0.0001
Triglyceride (mg/dL)	194.1 ± 15.1	171.7 ± 4.5	174.6 ± 11.1	0.5907	119.8 ± 3.7	129.1 ± 2.8	126.6 ± 6.1	0.0008
Macronutrient intake (mean ± SE)								
Energy (kcal)	2024 ± 34.6	2160.6 ± 16.3	2142.6 ± 46.7	<0.0001	1846.7 ± 26.8	1832.7 ± 11.5	1928.4 ± 36.4	<0.0001
Carbohydrate(g)	214.4 ± 7.9	322.7 ± 2.6	373.3 ± 8.0	<0.0001	188.8 ± 7.3	271.1 ± 1.8	338.7 ± 6.6	<0.0001
Protein (g)	78.3 ± 2.8	71.3 ± 0.6	64.9 ± 1.5	<0.0001	71.9 ± 2.4	59.9 ± 0.4	57.3 ± 1.1	<0.0001
Fat (g)	86.4 ± 3.4	65.0 ± 0.6	41.6 ± 1.0	<0.0001	81.1 ± 3.1	56.7 ± 0.4	37.7 ± 0.7	<0.0001
% Energy from								
Carbohydrate	42.2 ± 0.6	59.8 ± 0.2	69.8 ± 0.3	<0.0001	40.7 ± 0.6	59.8 ± 0.1	70.3 ± 0.2	<0.0001
Protein	15.6 ± 0.4	13.3 ± 0.1	12.2 ± 0.1	<0.0001	15.9 ± 0.4	13.2 ± 0.1	12.0 ± 0.1	<0.0001
Fat	38.3 ± 0.6	27.0 ± 0.1	17.5 ± 0.2	<0.0001	39.3 ± 0.5	27.8 ± 0.2	17.6 ± 0.1	<0.0001
EER%	72.0 ± 3.6	83.3 ± 0.8	84.9 ± 2.3	<0.0001	95.1 ± 3.6	97.6 ± 0.7	103.9 ± 2.1	<0.0001

**Abbreviations:** BMI, body mass index; DBP, diastolic blood pressure; HDL, high-density lipoprotein; LDL, low-density lipoprotein; SBP, systolic blood pressure; EER, estimated energy requirement. ^a^ Based on *χ*^2^ test for categorical variables and ANOVA for continuous variables.

**Table 3 nutrients-13-01834-t003:** Prevalence and multivariable-adjusted odds ratios for MetS among participants according to carbohydrate and fat intake.

		Men			Women	
	LCHF ^a^	MCF ^b^	HCLF ^c^	LCHF	MCF	HCLF
	(N = 330)	(N = 1410)	(N = 224)	(N = 797)	(N = 2907)	(N = 355)
Increased waist circumference						
Prevalence (%)	55.8	47.28	53.55	63.86	67.13	70.38
OR (95% CI)	1.67 (0.95–2.97)	1.38 (0.93–2.04)	1.00	0.85 (0.54–1.34)	0.75 (0.50–1.13)	1.00
Elevated blood pressure						
Prevalence (%)	25.01	28.99	30.31	10.70	13.72	24.97
OR (95% CI)	0.87 (0.50–1.55)	0.94 (0.61–1.44)	1.00	0.34 (0.19–0.59)	0.50 (0.32–0.79)	1.00
Reduced HDL cholesterol						
Prevalence (%)	50.44	42.25	50.73	59.73	65.34	66.13
OR (95% CI)	1.39 (0.88–2.44)	1.23 (0.79–1.91)	1.00	0.87 (0.57–1.33)	1.00 (0.70–1.43)	1.00
Elevated triglycerides						
Prevalence (%)	48.18	44.63	44.53	24.17	26.87	26.45
OR (95% CI)	1.10 (0.61–1.85)	0.81 (0.52–1.26)	1.00	0.97 (0.61–1.56)	1.12 (0.75–1.67)	1.00
Elevated fasting glucose						
Prevalence (%)	14.66	15.50	23.96	14.23	13.95	24.39
OR (95% CI)	0.82 (0.43–1.55)	0.69 (0.40–1.17)	1.00	0.68 (0.40–1.16)	0.58 (0.37–0.91)	1.00
Metabolic syndrome						
Prevalence (%)	37.03	36.80	34.26	27.00	28.63	38.50
OR (95% CI)	1.17 (0.62–2.18)	0.94 (0.57–1.54)	1.00	0.77 (0.46–1.29)	0.71 (0.47–1.07)	1.00

**Abbreviations:** OR, odd ratio; CI, confidence interval; HDL, high-density lipoprotein. ^a^ Carbohydrates (<45%); Fat (>30%). ^b^ Carbohydrates (55–65%); Fat (20–30%). ^c^ Carbohydrates (>65%); Fat (<20%). All values accounted for the complex sampling design effect of the national surveys using PROC SURVEY procedure. Ethnicity, family economic status, education level, elevation, relative humidity, BMI (except for the model of waist circumference), and total energy intake were adjusted for multiple logistic regression.

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
