# Peer review of "Association of Carbohydrate and Fat Intake with Prevalence of Metabolic Syndrome Can Be Modified by Physical Activity and Physical Environment in Ecuadorian Adults: The ENSANUT-ECU Study"

_nutrients, 2021, doi:10.3390/nu13061834_

Round 1

Reviewer 1 Report

The paper from Christian F. Juna et al describes the effect of physical activity and dietary abitudes on the prevalence of metabolic syndrome . In particular, they  utilized data from the Ecuador National Health and Nutrition Survey 2012 (ENSANUT-ECU) in which a logistic regression was used to determine the association of MetS with status of carbohydrate and fat intake.

Data are novel and provide useful information in this growing area of interest.

Author Response

  • We sincerely appreciate your valuable time to provide useful comments and feedback.

Reviewer 2 Report

This is an interesting paper concerning the effect of food intake as a combination of carbohydrates and fat on MetS. The authors also evaluated the association of other factors such as physical activity, environment, and food intake on MetS. All subjects are stratified by sex and status of carbohydrate and fat intake. The authors used chi-square to investigate the contributing factors on MetS, and then adopted linear regression to adjust confounding variables. Also, the multiple logistic regression was performed after adjusting confounding variables to determine the association of MetS with the status of carbohydrate in food intake using high fat low carbohydrate as a reference. Furthermore, the adjusted odds ratios of MetS according to diet, lifestyle, and physical environment were determined.  

This is a well-organized paper, which provides information about the association of carbohydrate and fat intake with the prevalence of MetS in Ecuadorian adults, which can serve as a reference to offer MetS-prevention education for people from countries with similar income. Some minor questions are as the followings.

  1. The protein content in different diet groups varies and exhibit significant difference. Please discuss if protein, instead of carbohydrate or fat, contributes to MetS. 
  2. The definition of LCHF indicates a low percentage of carbohydrates and a high percentage of fat as sources of energy. However, LCHF is not necessarily equal to low amounts of carbohydrates and high amounts of fat; it could be high or low. In other words, some LCHF subjects might consume a high amount of carbohydrates, which might interfere with the analysis. Please discuss whether the definition regarding the status of carbohydrates and fat in this study is suitable for measuring the association of diet and MetS. 

Author Response

General evaluation: This is a well-organized paper, which provides information about the association of carbohydrate and fat intake with the prevalence of MetS in Ecuadorian adults, which can serve as a reference to offer MetS-prevention education for people from countries with similar income.

  • We sincerely appreciate your valuable time to provide useful comments and feedback. We carefully considered your opinion and revised the manuscript based on your comments. Our responses to your comments are as follows.

Specific evaluation:

The protein content in different diet groups varies and exhibit significant difference. Please discuss if protein, instead of carbohydrate or fat, contributes to MetS.

  • We added the possible role of protein intake in the Discussion part as the reviewer suggested.

(Page 10, line 298-304) “In addition, protein content exhibited significant differences among diet groups in this study; an increased intake of protein has been reported to have a greater satiety effect than intake of carbohydrates, helping control hunger between meals and regulate the processes of energy expenditure, thermogenesis and glucose metabolism, which intervene in the pathophysiology of MetS [51, 52]. The association between protein intake and MetS is still unclear, as some studies have not found associations [53-55] while others have found that low protein intake could be a risk factor for MetS components [56].”

The definition of LCHF indicates a low percentage of carbohydrates and a high percentage of fat as sources of energy. However, LCHF is not necessarily equal to low amounts of carbohydrates and high amounts of fat; it could be high or low. In other words, some LCHF subjects might consume a high amount of carbohydrates, which might interfere with the analysis. Please discuss whether the definition regarding the status of carbohydrates and fat in this study is suitable for measuring the association of diet and MetS.

  • We appreciate this critical point. Since we could not get information on the actual types of carbohydrates and fat, we revised and added this point as another limitations of the study.

(Page 11, line 338-343) “Second, data on dietary intake from the 24 h dietary recall might not represent the usual food intake of participants. However, this study included a nation-wide representative sample of participants. Third, we could not estimate the differences in carbohydrate quality and fatty acid composition in diet groups due to the lack of information. Therefore, more studies are needed to clarify their associations with MetS and its components.”

  • We also performed a separate analysis and adjusted the model for % of energy from macronutrients and we did not find any significant difference from the first model (data not shown).

Reviewer 3 Report

The manuscript submitted for review is a valuable work on interactive associations among carbohydrate and fat intake, physical environment, lifestyle, and MetS among Ecuadorian adults. The scale of the study deserves special attention: it covered 6,023 participants.

I have some minor comments

  • In abrtact in found „participants aged 20 to 60 years”, but in introduction you stated „aged from 20 to 64 years”
  • Page 3, line 95: should be 6,023 (not 6023)
  • Page 4, line 143-148: the criteria for classification for MetS are described correctly, but many studies emphasize that also the pharmacotherapy used for controling the symptom (for example drugs to lower blood pressure or glucose levels) is a factor that prompts the inclusion of a given symptom as MetS +
  • Page 4, line 143-148: In the diagnosis of MetS, the principle, which is often use, is that we diagnose MetS when there are 3 out of 5 diagnostic criteria - I think it is worth supplementing this information

Author Response

General evaluation: The manuscript submitted for review is a valuable work on interactive associations among carbohydrate and fat intake, physical environment, lifestyle, and MetS among Ecuadorian adults. The scale of the study deserves special attention: it covered 6,023 participants.

  • We sincerely appreciate your valuable time to provide useful comments and feedback. We carefully considered your opinion and revised the manuscript based on your comments. Our responses to your comments are as follows.

Specific evaluation:

In abstract in found “participants aged 20 to 60 years”, but in introduction you stated “aged from 20 to 64 years”

  • We corrected the error we made.

(Page 3, line 93) “Participants in this study were Ecuadorian adults from ages 20 to 60 years…”

Page 3, line 95: should be 6,023 (not 6023)

  • We corrected the error we made.

(Page 3, line 97) “Thus, the final 6,023 Ecuadorian adults…”

Page 4, line 143-148: the criteria for classification for MetS are described correctly, but many studies emphasize that also the pharmacotherapy used for controlling the symptom (for example drugs to lower blood pressure or glucose levels) is a factor that prompts the inclusion of a given symptom as MetS +

  • We sincerely appreciate your useful comment; we will consider including this criterion in future studies.

Page 4, line 143-148: In the diagnosis of MetS, the principle, which is often use, is that we diagnose MetS when there are 3 out of 5 diagnostic criteria - I think it is worth supplementing this information

  • We modified the statement according to your suggestion.

(Page 4, line 144-145) “The subjects with MetS were identified when they had three or more of the following components:”

Reviewer 4 Report

The paper of Juna et al. try to study whether there is an   interaction between carbohydrate and fat intakes, lifestyles, and physical environment with MetS.

The authors report is detailed indeed, however this is one of its weaknesses. It was not easy to understand the whole picture and to come with a comprehensive conclusion. Moreover, I found the results inconsistent and confusing.

It seems that Carb's consumption is dependent with the socioeconomic status and this is the reason for the differences with living height? One should be familiar with Ecuador geography which I'm not.

In addition, there weren't significant anthropometric differences between groups and no difference with the prevalence of MeS, despite the different diet habits….

In women- low Carbs was associates with lower SBP but glucose was lower only in the MCF group?

Low Carbs was associate with higher waist circumference but the Tg were lower in this group.

I can show more examples but the bottom line is clear- I don’t think that there is important new scientific medical knowledge that is important internationally.

Author Response

General evaluation: This interesting study has examined the associations of relative humidity and lifestyle factors with metabolic syndrome in Ecuador. The findings are intriguing.

We sincerely appreciate your valuable time to provide useful comments and feedback. We carefully considered your opinion and revised the manuscript based on your comments and recommendations. Our responses to your comments are as follows.

Specific evaluation:

The authors report is detailed indeed, however this is one of its weaknesses. It was not easy to understand the whole picture and to come with a comprehensive conclusion. Moreover, I found the results inconsistent and confusing.

  1. It seems that Carb's consumption is dependent with the socioeconomic status and this is the reason for the differences with living height?

  • We have adjusted for socioeconomic status for our previous regression models; thus, our models have already considered this factor.

  • We also adjusted the model for geographic regions; however, there was no significant differences (data not shown). Furthermore, we included some statement of geographic regions of Ecuador in the Discussion section. Ecuador is 1 of the 17 megadiverse countries of the world, it has a variety of microclimates and a wide gap of elevation of residence, resulting in varied lifestyles among the population. In consequence, we consider that countries with similar characteristics could benefit from the findings of the present study.

(Page 10, line 307-311) “Ecuador is a megadiverse country composed by four different geographical regions (the Coast, the Galapagos Islands, the Amazon, and the Andean region); its population resides in an elevation rage of 0 masl to 3900 masl. In addition, Ecuador has 11 different types of microclimates ranging from Tropical to Oceanic and is located on the equatorial line, thus producing little seasonality.”

  1. In addition, there weren't significant anthropometric differences between groups and no difference with the prevalence of MetS, despite the different diet habits

We had our models rechecked and retested the difference of variables among diet groups using ANOVA test, and some anthropometric/biochemical measurement were significantly different by diet groups. We edited our statement about this change in the Results section as well as Table 2.   

(Pages 5-6, line 184-192) “Table 2 shows anthropometric and biochemical measurements for MetS components and macronutrient intakes. Blood pressure, total cholesterol, triglycerides, and energy intake were higher in men, but BMI and EER (%) were higher in women. Significant differences between diet groups were observed in BMI, systolic blood pressure, diastolic blood pressure, and HDL cholesterol in both sexes (p<0.05), while in men, waist circumference was significantly different (p<0.0001) and in women, fasting glucose, total cholesterol, LDL cholesterol and triglycerides (p=0.0126, p=0.0002, p<0.0001, p=0.0008, respectively). Moreover, men and women with HCLF diets showed a higher energy intake and EER (%) (p<0.001, both).”

  1. In women- low Carbs was associates with lower SBP but glucose was lower only in the MCF group?

  • The associations of MetS (including its components) and macronutrient intake are still imprecise. We found these associations in Ecuadorian women; however, more research may be needed to clarify this association in other populations and more detailed analyses regarding carbohydrate and fat composition with MetS. Therefore, we described this point as a limitation of the study.

(Page 11, line 335-338) “First, the cross-sectional design of the ENSANUT-ECU data made it difficult to identify the causal relationship between carbohydrate and fat intake with MetS”

(Page 11, line 340-343) “Third, we could not estimate the differences in carbohydrate quality and fatty acid com-position in diet groups due to the lack of information. Therefore, more studies are needed to clarify their associations with MetS and its components.”

  1. Low Carbs was associated with higher waist circumference but the Tg were lower in this group.

LCHF diet was associated with higher waist circumference only in men, not in the women’s group, where LCHF diet had the lowest levels of triglycerides. However, we further analyzed multiple regression models including health related lifestyles and physical environment to determine if there were potential interactive associations among variables. We found that not performing physical activity and living in high relative humidity induced higher prevalence of increased waist circumferences independently of the diet type.